Factor structure of the Positive and Negative Affect Schedule (PANAS) in adult women with fibromyalgia from Southern Spain: the al-Ándalus project

http://orcid.org/0000-0003-2960-4142 Estévez-López Fernando 1 2 festevez@ugr.es
Pulido-Martos Manuel 3
Armitage Christopher J. 4
Wearden Alison 4
Álvarez-Gallardo Inmaculada C. 1
Arrayás-Grajera Manuel Javier 5
Girela-Rejón María J. 1
Carbonell-Baeza Ana 6
Aparicio Virginia A. 7 8
Geenen Rinie 2
Delgado-Fernández Manuel 1
http://orcid.org/0000-0001-8655-9857 Segura-Jiménez Víctor 1 6
1 Department of Physical Education and Sport, Faculty of Sport Sciences, University of Granada , Granada , Spain
2 Department of Psychology, Faculty of Social and Behavioural Sciences, Utrecht University , Utrecht , The Netherlands
3 Department of Psychology, Faculty of Humanities and Education Sciences, University of Jaén , Jaén , Spain
4 Manchester Centre for Health Psychology, School of Psychological Sciences, Manchester Academic Health Science Centre, University of Manchester , Manchester , United Kingdom
5 Department of Physical Education, Music and Fine Arts, Faculty of Education Sciences, University of Huelva , Huelva , Spain
6 Department of Physical Education, Faculty of Education Sciences, University of Cádiz , Cádiz , Spain
7 Department of Physiology and Institute of Nutrition and Food Technology, Faculty of Pharmacy, University of Granada , Granada , Spain
8 Department of Public and Occupational Health, EMGO+ Institute for Health and Care Research, VU University Medical Care , Amsterdam , The Netherlands
Ozakinci Gozde
Electronic publication date: 2016 Mar 24
Publication date: 2016
Volume: 4
Electronic Location ID: e1822
Received 2015 Dec 15; Accepted 2016 Feb 26
Copyright: © 2016 Estévez-López et al.
Copyright year: 2016
Copyright holder: Estévez-López et al.
License: This is an open access article distributed under the terms of the Creative Commons Attribution License, which permits unrestricted use, distribution, reproduction and adaptation in any medium and for any purpose provided that it is properly attributed. For attribution, the original author(s), title, publication source (PeerJ) and either DOI or URL of the article must be cited.
License URL: https://creativecommons.org/licenses/by/4.0/

Keywords: Affectivity, Chronic pain, Confirmatory factor analysis, Dimensional structure, Emotion, Mood, Positive and Negative Affect Schedule, Psychometrics

Funding: Spanish Ministry of Economy and Competitiveness I+D+i DEP2010-15639 and I+D+i DEP2013-40908-R Consejería de Turismo, Comercio y Deporte CTCD-201000019242-TRA FE-L BES-2014-067612 ICA-G BES-2011-047133 This study was funded by the Spanish Ministry of Economy and Competitiveness [I+D+i DEP2010-15639 and I+D+i DEP2013-40908-R], the Consejeria de Turismo, Comercio y Deporte (CTCD-201000019242-TRA), the Andalusia Institute of Sport, the Center of Initiatives and Cooperation to the Development (CICODE, University of Granada), and the Andalusian Federation of people with fibromyalgia, chronic fatigue, and multiple, chemical sensitivity (Alba Andalucía). FE-L [Grant number: BES-2014-067612] and ICA-G [Grant number: BES-2011-047133] were supported by Grants from the Spanish Ministry of Economy and Competitiveness. VAA was supported by the Andalucía Talent Hub Program launched by the Andalusian Knowledge Agency, co-funded by the European Union’s Seventh Framework Program, Marie Skłdowska-Curie actions (COFUND–Grant Agreement no 291780) and the Ministry of Economy, Innovation, Science and Employment of the Junta de Andalucía. The funders had no role in study design, data collection and analysis, decision to publish, or preparation of the manuscript.

==============================
Background: Fibromyalgia is a syndrome characterized by the presence of widespread chronic pain. People with fibromyalgia report lower levels of Positive Affect and higher levels of Negative Affect than non-fibromyalgia peers. The Positive and Negative Affect Schedule (PANAS)–a widely used questionnaire to assess two core domains of affect; namely ‘Positive Affect’ and ‘Negative Affect’ –has a controversial factor structure varying across studies. The internal structure of a measurement instrument has an impact on the meaning and validity of its score. Therefore, the aim of the present study was to assess the structural construct validity of the PANAS in adult women with fibromyalgia. Methods: This population-based cross-sectional study included 442 adult women with fibromyalgia (age: 51.3 ± 7.4 years old) from Andalusia (Southern Spain). Confirmatory factor analyses were conducted to test the factor structure of the PANAS. Results: A structure with two correlated factors (Positive Affect and Negative Affect) obtained the best fit; S-B χ2 = 288.49, df = 155, p < .001; RMSEA = .04; 90% CI of RMSEA = (.036, .052); the best fit SRMR = .05; CFI = .96; CAIC = −810.66, respectively. Conclusions: The present study demonstrates that both Positive Affect and Negative Affect are core dimensions of affect in adult women with fibromyalgia. A structure with two correlated factors of the PANAS emerged from our sample of women with fibromyalgia from Andalusia (Southern Spain). In this model, the amount of variance shared by Positive Affect and Negative Affect was small. Therefore, our findings support to use and interpret the Positive Affect and Negative Affect subscales of the PANAS as separate factors that are associated but distinctive as well.

Introduction

The assessment of affect–subjectively experienced feeling or emotion–is important because affect is a primary cause and consequence of relevant phenomena such as coping, symptoms, social activity and satisfaction among others (Watson, Clark & Tellegen, 1988). The Positive and Negative Affect Schedule (PANAS) (Watson, Clark & Tellegen, 1988) is probably the most extensively-used instrument to measure affect (Buz et al., 2015). Overall, the PANAS often shows appropriate psychometric properties including internal consistency and test-retest reliabilities as well as convergent and discriminant validities (Watson, Clark & Tellegen, 1988). However, the structural construct validity of the PANAS is controversial; its factor structure varies across studies.

The original authors (Watson, Clark & Tellegen, 1988), in an adult sample and using Exploratory Factor Analyses (EFA), indicated that the PANAS consists in a structure with two largely uncorrelated factors; namely ‘Positive Affect’ and ‘Negative Affect.’ Subsequent research has been conducted across different populations and languages of the scale, using state or trait time-frame directions (i.e., about current or general feelings, respectively), and different statistical techniques (e.g., EFA or Confirmatory Factor Analyses (CFA)). However, the structure that emerged as the most appropriate in each study did not seem to depend on the above-mentioned issues. For instance, different structures have emerged in two studies conducted among university students from Spain, using the Spanish version of the PANAS with trait time-frame directions, and with CFA (Sandin et al., 1999; Ortuño-Sierra et al., 2015).

Several techniques are available to analyse the factor structure of a questionnaire. In the early phase of the PANAS, EFA were used (Watson, Clark & Tellegen, 1988; Mehrabian, 1997; Killgore, 2000). In the next step, CFA allowed estimation and testing of a hypothesized model based on previous literature, of correlated uniqueness terms, factor variances, factor covariances, comparison of competing models and selection of the best fitting model (Lloret-Segura et al., 2014). CFA and Item-Response Theory (ITR) are often considered as complementary approaches; CFA is suggested to be a more appropriate approach when analysing multidimensional models (Wang, 2005) while ITR is better suited for testing equivalence of item parameters (Meade, 2004). Most studies tested the factor structure of the PANAS using CFA (Leue & Beauducel, 2011). Therefore, and for the sake of clarity, only CFA literature related with the PANAS is mentioned hereinafter.

After the original work (Watson, Clark & Tellegen, 1988), most of research has provided further evidence of a structure with two factors of the PANAS. However, the correlation between Positive Affect and Negative Affect is still under debate, with studies supporting (Crocker, 1997; Sandin et al., 1999; Tuccitto, Giacobbi & Leite, 2009) or rejecting (Crawford & Henry, 2004; Lim et al., 2010) the original relatively independent relation between the two dimensions of affect. A few studies reported more complex structures (Leue & Beauducel, 2011; Ortuño-Sierra et al., 2015) or did not obtain a good fit to the data with any of the models that were assessed (Melvin & Molloy, 2000; Molloy, Pallant & Kantas, 2001; Beck et al., 2003), including the original structure.

To date, most studies have examined the factor structure of the PANAS among student and community samples, while affect also plays an important role in clinical and forensic samples (Leue & Beauducel, 2011). Our review of the literature elicited just two studies (Beck et al., 2003; Lim et al., 2010) that have addressed the factor structure of the PANAS with CFA in clinical samples. Both studies failed to replicate the original structure. Beck et al. (2003) showed a poor fit to the original structure in a sample of older adults with generalized anxiety disorder. Lim et al. (2010) found a structure with two correlated factors in a sample of psychiatric patients. Although two other studies did include people with clinical disorders in their samples, unfortunately the CFA was conducted in the whole sample instead of in the patient subsample (Gyollai et al., 2011; Leue & Beauducel, 2011). To know the nature of the association between the two dimensions of the PANAS in clinical populations is important because it shows how the PANAS should be used and interpreted in clinical settings.

Fibromyalgia is a chronic pain condition characterised by high sensitivity to painful stimuli and lowered pain threshold; namely ‘hyperalgesia’ and ‘allodynia,’ respectively (Kosek, Ekholm & Hansson, 1996). Recent research has focussed on the emotional lives of people with fibromyalgia and its associated factors (Córdoba-Torrecilla et al., in press; Soriano-Maldonado et al., 2015a; Soriano-Maldonado et al., 2015b) showing that they have lower levels of Positive Affect and higher levels of Negative Affect than control peers (Hassett et al., 2008; van Middendorp et al., 2010). Additionally, Negative Affect and Positive Affect are associated to fibromyalgia severity, fatigue and pain (van Middendorp et al., 2008; Estévez-López et al., 2015).

The Dynamic Model of Affect (Reich, Zautra & Potter, 2001; Reich, Zautra & Davis, 2003) posits that the structure of affect varies from ordinary circumstances–in which people experience low stress–to stressful conditions. When stress is low, people are able to focus their cognitive resources–and, therefore, process more information–on their affective complexity. However, when stress is high, there is a competition for cognitive resources implying a depletion of cognitive resources that are available for processing affective information. Consequently, when stress is high, Positive Affect and Negative Affect become increasingly and inversely correlated. Evidence from chronic pain populations has corroborated the assumption of less differentiation of Positive Affect and Negative Affect when people experience increased levels of stress (Zautra et al., 2001; Davis, Zautra & Smith, 2004; Zautra et al., 2007; Zautra et al., 2005).

The reliable and valid measurement of affect in fibromyalgia is important because according to the Dynamic Model of Affect-in this group that is characterized by pain and stress, Positive Affect and Negative Affect may correlate more than in the general population (Reich, Zautra & Potter, 2001; Reich, Zautra & Davis, 2003). Thus, it is possible that the two factors of the PANAS may correlate more clearly in fibromyalgia than in pain-free populations. To the best of our knowledge, no studies have yet assessed the internal structure of the PANAS in people with fibromyalgia. Given that the PANAS may be sensitive to sampling (Crawford & Henry, 2004; Tuccitto, Giacobbi & Leite, 2009) and is frequently used in studies of fibromyalgia (Zautra, Johnson & Davis, 2005; Sturgeon, Zautra & Arewasikporn, 2014; Estévez-López et al., 2015), it seems necessary to examine its structure in this particular population to properly use and interpret the PANAS score. Therefore, the aim of the present study was to assess the structural construct validity of the PANAS in adult women with fibromyalgia from Andalusia (Southern Spain) using CFA.

Methods

Participants

We focused to recruit a representative sample of people with fibromyalgia from Andalusia (Southern Spain). To achieve this representativeness, calculations of sample size were conducted as elsewhere (Segura-Jiménez et al., 2015). Through the Andalusian Federation of Fibromyalgia, we contacted the main local associations of people with fibromyalgia in all provinces of Andalusia (Southern Spain)–i.e., those with the largest number of people. All interested participants (n = 616) received an invitation to participate in this study. Potentially eligible participants who were interested in participating in the study attended a meeting where we provided information about the study aims and procedures. Those taking part in the study signed a written informed consent form.

Inclusion criteria for the present study were: (i) to be an adult woman (aged 18–65 years old), (ii) to have a medical diagnosis of fibromyalgia by a rheumatologist (participants were requested to provide their medical records to confirm their diagnosis) and (iii) to meet the American College of Rheumatology 1990 fibromyalgia criteria (Wolfe et al., 1990), namely, widespread pain for more than 3 months, and pain with 4 kg/cm of pressure reported for 11 or more of 18 tender points. Additionally, participants with acute or terminal illness or severe cognitive impairment (i.e., Mini Mental State Examination (MMSE) < 10) were excluded.

All participants were assessed by the same research group to reduce inter-examiner error. The research protocol was reviewed and approved by the Ethics Committee of the Hospital Virgen de las Nieves (Granada, Spain); Registration number: 15/11/2013-N72.

Measures

Sociodemographic data

Sociodemographic information was recorded using a self-report instrument that included date of birth, time since fibromyalgia diagnosis, time from first symptoms until fibromyalgia diagnosis, and marital, educational, and current occupational status. Additionally, participants were asked: ‘Have you ever been diagnosed with an acute or terminal illness?’ to assess one of the exclusion criteria.

Tenderness

The 18 tender points were assessed in accordance to the 1990 American College of Rheumatology criteria (Wolfe et al., 1990; Segura-Jiménez et al., 2014) for diagnosis and classification of fibromyalgia. A standard pressure algometer (FPK 20; Wagner Instruments, Greenwich, CT, USA) was used. Total tender points count was recorded for each participant.

The MMSE

The MMSE (Folstein, Folstein & McHugh, 1975; Lobo et al., 1979) was used to evaluate severe cognitive impairment-i.e., to assess one of the exclusion criteria.

The PANAS

The PANAS (Watson, Clark & Tellegen, 1988; Sandin et al., 1999) is a questionnaire designed to assess Positive and Negative Affect. This questionnaire has 20 items, 10 to Positive Affect (e.g., enthusiastic) and 10 to Negative Affect (e.g., scared). Participants responded to each item on a 5-point Likert-type scale: 1) very slightly or not at all, 2) a little, 3) moderately, 4) quite a bit, and 5) extremely. The time-frame adopted was in ‘general.’ The items of the PANAS are: Interested, distressed, excited, upset, strong, guilty, scared, hostile, enthusiastic, proud, irritable, alert, ashamed, inspired, nervous, determined, attentive, jittery, active, and afraid. The scores range is 10–50 for both Positive Affect and Negative Affect.

Procedure

During a single evaluation session participants filled out sociodemographic data and answered questions of the MMSE. Then, a physical examination was conducted to assess tender points. Finally, the PANAS was completed at home the following day and delivered to the research team two days after the evaluation session.

Statistical analysis

Preliminary analysis

The Statistical Package for Social Sciences (IBM SPSS for Mac, version 20.0; Armonk, NY, USA) was used to: (a) examine the pattern of missing data and (b) obtain descriptive data. Structural Equation Modeling Software (EQS 6.1 for Windows; Encino, CA, USA) was used to evaluate potential violations of normality assumptions. Mardia’s coefficient was calculated to check multivariate normality.

Measurement model

Factor structure and fit models were obtained with CFA using EQS 6.1. We chose those goodness-of-fit indices that were the most insensitive to sample size, model misspecification and parameter estimates (Hooper, Coughlan & Mullen, 2008). Specifically, we based our decisions on the following absolute fit indices: (i) the Satorra-Bentler scaled χ2 statistic (S-B χ2) and its degree of freedom (df) and p values, (ii) the Root Mean Square Error of Approximation (RMSEA) with the 90% confidence interval (90% CI) (Steiger, 1990), and (iii) the Standardized Root Mean Squared Residual (SRMR), (iv) the Comparative Fix Index (CFI), as an incremental fit index, and (v) the Consistent version of Akaike’s Information Criterion (CAIC) (Bozdogan, 1987), as a parsimony fit index to compare the fit of non-nested models. As some indicators were non-normally distributed, we decided to use Robust Maximum Likelihood (MLM) analyses. An acceptable model fit was defined by: S-B χ2 (lower values of χ2 indicate a good fit if p > .05) (Bollen, 1989); RMSEA (values ≤ .08 indicate acceptable; < .05 indicate relatively good fit) (Browne & Cudeck, 1992); SRMR (values ≤ .05 indicate well-fitting model); CFI (values .90–.94 indicate acceptable fit; ≥ .95 relatively good fit) (Hu & Bentler, 1999); and a small value of CAIC when comparisons among non-nested model are done (Hu, Bentler & Hoyle, 1995). Although, the comparison between nested models assesses by S-B χ2 is not distributed like χ2 (Satorra, 2000), it has been suggested that a scaled difference χ2 can be used to compare S-B χ2 from nested models (Satorra & Bentler, 2001).

CFA-based scale reliability

When correlated error terms are allowed, conventional estimates of reliability (e.g., Cronbach’s Alpha) may be biased (Hankins, 2008). Therefore, internal consistency of Positive Affect and Negative Affect were computed with composite reliability (ρ) (Raykov, 2001; Raykov, 2004). A ρ > .70 was considered as a minimum acceptable cut-off value, which is in line with the interpretation of Cronbach’s alpha (Nunnally & Bernstein, 1994).

Parametrization of CFA models

CFA was used to test the factor structure of the PANAS in the present dataset against all previous models that have assessed the factor structure of the PANAS using CFA.

Model 1 was a general model, with the 20 items hypothesized to load onto a single factor.

Model 2a consisted of two uncorrelated factors, Positive Affect and Negative Affect, proposed originally by Watson, Clark & Tellegen (1988) and supported also in further studies (Crocker, 1997; Schmukle, Egloff & Burns, 2002; Terracciano, McCrae, & Costa, 2003; Tuccitto, Giacobbi & Leite, 2009; Gyollai et al., 2011; Buz et al., 2015). Model 2a(1) was the same as Model 2a, except that error terms from some items was permitted to intercorrelate, following the Zevon & Tellegen’s (1982) mood subcategories.1 Model 2a(2) was the same as Model 2a, but allowing to covary the error terms of the items: hostile and irritable; ashamed and guilty; scared and afraid; nervous and jittery; and ashamed and determined (Sandin et al., 1999). Model 2b was a structure with two correlated factors (Schmukle, Egloff & Burns, 2002). Model 2b(1) was the same as Model 2b, except that error terms from some items was permitted to intercorrelate, following Zevon & Tellegen’s (1982) mood subcategories (Crawford & Henry, 2004). Model 2b(2) was the same as Model 2b, with different error terms correlated: guilty and ashamed; scared and afraid; attentive and alert; interested and excited; and inspired and strong (Lim et al., 2010). Model 2b(3) was proposed by the authors of the present work, based on the results of Lagrange Multiplier (LM) test, and was the same as Model 2b(1) adding a correlation of the error terms of the items distressed and nervous.

Model 3a was a structure with three first-order uncorrelated factors: (i) the Positive Affect factor with the original 10 items, (ii) the Negative Affect-1 factor with eight of the original 10 items, without guilty and ashamed, and (iii) the Negative Affect-2 factor with only two items: guilty and ashamed (Beck et al., 2003). Model 3a(1) was a structure with three first-order uncorrelated factors: (i) the Positive Affect factor with the original 10 items, (ii) the Afraid factor with 5 items: distressed, scared, nervous, jittery, and afraid, and (iii) the Upset factor with 5 items: upset, guilty, hostile, irritable, and ashamed (Gaudreau, Sanchez & Blondin, 2006; Ortuño-Sierra et al., 2015). Model 3a(2) was the same as Model 3a(1), except that cross-loading to some items were allowed: nervous and jittery loaded on Positive Affect, excited loaded on Afraid (see Gaudreau, Sanchez & Blondin (2006), ‘calibration sample’). Model 3a(3) was the same as Model 3a(1), except that cross-loading to some items were allowed: active and alert (originally from Positive Affect), and hostile (originally from Upset) loaded on Afraid (see Gaudreau, Sanchez & Blondin (2006), ‘cross-validation sample’). Models 3b(1), 3b(2) and 3b(3) were the same as Models 3a(1), 3a(2) and 3a(3), respectively, except that it was a correlated three first-order factor structure (Leue & Beauducel, 2011).

Model 4 was a bifactor model with Positive Affect and Negative Affect as independent factors as well as an additional general one, Affective Polarity factor, with all 20 items of the questionnaire (Leue & Beauducel, 2011; Lopez-Gomez, Hervas & Vazquez, 2015).

Results

From 616 potentially eligible people with fibromyalgia, a number of participants were excluded because they were men (n = 21), older than 65 years old (n = 25), did not provide evidence of a medical diagnosis of fibromyalgia (n = 39), did not meet the diagnosis criteria (n = 83), or showed severely impaired cognitive performance (n = 1). Five participants failed to complete the PANAS and their data were subsequently excluded leaving a final sample of 442 people for analysis.

Preliminary analyses

Characteristics of the sample are displayed in Table 1. The mean for Positive Affect was 23.06 (SD = 6.79) and the mean for Negative Affect was 24.06 (SD = 8.42).

Table 1 Sociodemographic and clinical characteristics of the study sample of adult women with fibromyalgia (n = 442).

Characteristic	Value	
Demographic	
 Age (years old), mean (SD)	51.3 (7.42)	
 Marital status		
  Married	335 (75.8)	
  Single	36 (8.1)	
  Separated/Divorced/Widow	71 (16.1)	
Educational status	
 No studies	40 (9.0)	
 Primary school	213 (48.2)	
 Secondary school	126 (28.5)	
 University degree	63 (14.3)	
Current occupational status	
 Working full time	70 (15.8)	
 Working part time	46 (10.4)	
 Housewife	138 (31.2)	
 Student	5 (1.1)	
 Sick leave	32 (7.2)	
 Unemployed	81 (18.3)	
 Retired/pensioner/incapacity pension	70 (15.8)	
Fibromyalgia diagnosis	
 Time since diagnosis	
  Less than 1 year	29 (6.6)	
  Between 1 and 5 years	149 (33.7)	
  More than 5 years	251 (56.8)	
  No answer	13 (2.9)	
 Time from first symptoms until diagnosis	
  Less than 1 year	9 (2.0)	
  Between 1 and 5 years	214 (48.4)	
  More than 5 years	206 (46.6)	
  No answer	13 (2.9)	
 Total tender points count, mean (SD)	16.8 (1.94)	
Notes:

Values expressed as frequency (percentage) unless otherwise indicated.

SD, Standard Deviation.

Univariate normal analysis showed values between 0.04 and 2.01 for asymmetry, and between −1.02 and 2.01 for kurtosis, both within the recommended range (Chou & Bentler, 1995). Mardia’s coefficient was 117.71 and a critical proportion of 41.71, which showed that the data were not distributed normally (Bentler & Wu, 2005).

Confirmatory factor analyses

Although, based on CFI, the fit of various models would be considered adequate, a less parsimonious strategy based on the use of absolute misfit indices (Browne & Cudeck, 1992) like RMSEA and SRMR was established. When RMSEA and SRMR are used together, rejecting models with good fit is avoided (Hu & Bentler, 1998; Hu & Bentler, 1999).

The fit statistics for the CFA models are presented in Table 2. It can be seen that the general factor model (Model 1) showed a poor fit for all indices.

Table 2 Goodness of fit indices for the models assessed (n = 442).

	S-B χ2	df	RMSEA	90% CI of RMSEA	SRMR	CFI	CAIC	
Model 1	1400.09	170	.13	(.122, .134)	.14	.61	194.57	
Model 2a	637.74	170	.08	(.072, .085)	.13	.85	−567.88	
Model 2a(1)	361.23	157	.05	(.047, .062)	.13	.93	−752.10	
Model 2a(2)	430.57	165	.06	(.053, .067)	.13	.92	−739.50	
Model 2b	587.76	169	.07	(.068, .081)	.06	.87	−610.67	
Model 2b(1)	315.38	156	.05	(.040, .056)	.05	.95	−790.86	
Model 2b(2)	436.28	162	.06	(.055, .069)	.05	.91	−712.50	
Model 2b(3)	288.49	155	.04	(.036, .052)	.05	.96	−810.66	
Model 3a	739.13	170	.09	(.081, .093)	.17	.82	−466.39	
Model 3a(1)	778.58	170	.09	(.084, .096)	.20	.81	−426.94	
Model 3a(2)	763.07	167	.09	(.083, .096)	.19	.81	−421.18	
Model 3a(3)	768.58	167	.09	(.084, .097)	.20	.81	−415.67	
Model 3b(1)	548.02	169	.07	(.065, .078)	.13	.88	−650.41	
Model 3b(2)	534.04	166	.07	(.064, .078)	.12	.88	−643.12	
Model 3b(3)	533.20	166	.07	(.064, .077)	.13	.88	−643.95	
Model 4	498.69	150	.07	(.066, .080)	.13	.89	−565.00	
Note:

For all S-B χ2, p < .001. CAIC, Consistent version of Akaike’s Information Criterion; CFI, Comparative Fit Index; CI, Confidence Interval; df, degrees of freedom; RMSEA, Root Mean-Square Error of Approximation; SRMR, Standardized Root Mean-Square Residual; S-B χ2, Satorra–Bentler χ2 statistic.

Model 2a reflected the original hypothesis that the PANAS consists of a structure with two uncorrelated factors, Positive Affect and Negative Affect. The fit of this model was poor, though better when error terms were allowed to correlate following mood subcategories (Zevon & Tellegen, 1982) or the results of a Spanish adaptation (Sandin et al., 1999): Models 2a(1) and 2a(2), respectively. Finally, both models were rejected because values of SRMR (0.13) were inadequate.

Overall, better fit indices emerged when Positive Affect and Negative Affect were considered as correlated factors (Model 2b). Additionally, the fit of the models improved when the errors terms from some items were allowed to intercorrelate: Models 2b(1), 2b(2) and 2b(3). Specifically, Model 2b(1) followed mood subcategories (Zevon & Tellegen, 1982), Model 2b(2) was based on the results reported by Lim et al. (2010), and Model 2b(3) was a proposal by the authors of the present research based on the results of LM test. Specifically, Model 2b(3) added a new correlation of the error terms of the items distressed and nervous to the Model 2b(1). Model 2b(3), permitting these correlated errors, possessed markedly superior fit compared with their more constrained counterparts.

Models with a structure with three factors (i.e., 3a, 3a(1), 3a(2), 3a(3), 3b(1), 3b(2) and 3b(3)) or with the bifactor structure (Model 4) did not show an optimal fit to the data.

To select the best fitting model based on statistical grounds, comparisons between nested models were performed to assess the relative fit of each model that showed optimal goodness-of-fit indices to the data. Table 3 shows values of S-B χ2 per comparison of models. Models 2b(2) and 2b(3) obtained better fit values and differed statistically significant as compared to more restrictive models. The CAIC index was used to compare non-nested models and assess the parsimony of the models, taking into account the number of estimate parameters and the sample size. Model 2b(3) obtained a better value (i.e., a lower number) than Model 2b(2) (Table 2). Figure 1 illustrates Model 2b(3); i.e., the model that obtained the best fit indices: S-B χ2 = 288.49, df = 155, p < .001; RMSEA = .04; 90% CI of RMSEA = (.036, .052); SRMR = .05; CFI = .96; CAIC = −810.66.

Table 3 Comparison of the nested models using Satorra–Bentler χ2 statistic (n = 442).

	Model 2a(1)	Model 2a(2)	Model 2b(1)	Model 2b(3)	Model 2b(2)	
	Δ S-B χ2	Δdf	Δ S-B χ2	Δdf	Δ S-B χ2	Δdf	Δ S-B χ2	Δdf	Δ S-B χ2	Δdf	
Model 2a	219.03*	13	150.57*	5	258.57*	14	302.55*	15	181.02*	8	
Model 2b	–	–	92.44*	4	220.33*	13	243.63	14	121.10*	7	
Model 2a(1)	–	–	–	–	−83.18	1	228.11*	2	–	–	
Model 2b(1)	–	–	–	–	–	–	23.20*	1	–	–	
Notes:

* p < .01.

S-B χ2, Satorra–Bentler χ2 statistic; df, degrees of freedom.

Figure 1 Graphical representation of the correlated two-factor model of the PANAS, Model 2b(3); the factor loadings are standardized loadings.

CFA-based scale reliability

The values of ρ were .79 for the Positive Affect subscale and .76 for the Negative Affect subscale.

Discussion

Although the PANAS is extensively used in people with fibromyalgia, its structural construct validity has never been assessed in this population. This is an important omission because the internal structure of a measurement instrument impacts on the validity of its score interpretations (Messick, 1980). Thus, the aim of present research was to assess the fit of different hypothesized factor structures of the PANAS to the data from a sample of adult women with fibromyalgia from Andalusia (Southern Spain) using CFA. The results of previous CFA with the Spanish version of the PANAS showed a structure with two uncorrelated factors in women attending a program for the prevention of breast cancer (Joiner et al., 1997), university students (Sandin et al., 1999), and in older adults (Buz et al., 2015). These findings were congruent with the original proposed structure (Watson, Clark & Tellegen, 1988). However, another study also conducted among university students rejected such original structure and suggested a structure with three factors: Positive Affect, Negative Affect-Upset and Negative Affect-Afraid (Ortuño-Sierra et al., 2015). Recently, a bifactor structure has emerged from the Spanish general population (Lopez-Gomez, Hervas & Vazquez, 2015). Our current study provided an alternative model (i.e., a structure with two correlated factors) in which the two factors of the PANAS are associated but distinctive as well. This suggests that people with fibromyalgia may experience affect differently than non-fibromyalgia peers, which is in agreement with previous literature (Zautra, Johnson & Davis, 2005; Hassett et al., 2008; van Middendorp et al., 2008).

The best fit was obtained with the Model 2b(3): a structure with two correlated factors allowing intercorrelation of error terms from some items: attentive, interested, and alert; enthusiastic, excited, and inspired; proud and determined; strong and active; distressed and upset; hostile and irritable; scared and afraid; ashamed and guilty; nervous and jittery, and, distressed and nervous. Except the last one, all of them were based on mood subcategories (Zevon & Tellegen, 1982). Following a post-hoc modification suggested from our results, the error terms of the items distressed and nervous was allowed. In everyday Spanish language, distressed and nervous are often used interchangeably and are regarded as close synonyms. Thus, the correlation between the error terms of the items distressed and nervous was accepted given that it was justified and interpreted substantively (Jöreskog, 1967). The structure that emerged from the data is very close to the model that was obtained by Crawford & Henry (2004). Additionally, composite reliability of the Positive Affect and Negative Affect subscales showed an acceptable internal consistency of the PANAS; which is in line with Tuccitto, Giaboabbi & Leite (2009) who computed the composite reliability of Positive Affect and Negative Affect using data from previous studies (Lonigan et al., 1999; Crawford & Henry, 2004).

The present structure with two correlated factors has emerged in other investigations, even when they have used different type of samples and time-frame directions (Lonigan et al., 1999; Schmukle, Egloff & Burns, 2002; Terracciano, McCrae & Costa, 2003). It would be valuable to investigate the circumstances in which the present factor structure is replicated in future research. This is important because Watson, Clark & Tellegen (1988) argued that Positive Affect and Negative Affect reflect separate behavioural inhibition and behavioural engagement systems that the present study confirms by adding that there is an overlap between the two systems as well (Diener, Smith & Fujita, 1995; Tellegen, Watson & Clark, 1999).

The present findings are in support of the Dynamic Model of Affect (Reich, Zautra & Davis, 2003; Davis, Zautra & Smith, 2004; Zautra et al., 2007), which suggests that the relationship between Positive Affect and Negative Affect depends on the level of stress. It is plausible that in people with high levels of stress, the inverse correlation between Positive Affect and Negative Affect is strong, whereas this correlation is weaker in people with low stress levels. This could be an adaptive process whereby people with fibromyalgia more efficiently process both Positive Affect and Negative Affect information during stress episodes to allow simpler and more rapid information processing than in more relaxed situations.

Besides being bothered by widespread chronic pain and other symptoms that are uncertain and difficult to understand (Wright, Zautra & Going, 2008), in fibromyalgia, stress and distress levels are relatively high (Finan, Zautra & Davis, 2009), emotions are experienced more intensely (van Middendorp et al., 2008), and many people report to be confronted with socially invalidating responses (Kool et al., 2009; Kool et al., 2011; Kool et al., 2014). Experience sampling (Finan, Zautra & Davis, 2009) showed that people with fibromyalgia struggled to differentiate between Positive Affect and Negative Affect after a stressful stimulus. Moreover, people with fibromyalgia have greater difficulty differentiating between emotions (alexithymia) and more frequently use emotionally avoidant strategies (e.g., emotion suppression) than non-fibromyalgia peers (van Middendorp et al., 2008), which is considered maladaptive (Wiebe & Korbel, 2003; Geenen et al., 2012). The poor ability to discriminate between emotions and the use of avoidant emotion regulation responses might explain the structure with two correlated factors that emerged in the present study (van Middendorp et al., 2008; Lumley et al., 2011).

The present findings show that the fit of the model with a single factor (Model 1) and the fit of the most complex models (i.e., structures with three factors and the bifactor structure; Models 3a(1)–3b(3) and 4, respectively) were the poorest fitting structures. The models with structures with two uncorrelated factors were improved when correlation between error terms of some items was allowed (Models 2a(1) and 2a(2)). The highest fit indices were obtained with structures with two correlated factors and when correlation among error terms of some items was allowed (Models 2b(1)–2b(3)). The Model 2b(3) showed the best fit. Therefore, the present study demonstrates that both Positive Affect and Negative Affect are core dimensions of affect in adult women with fibromyalgia and highlights the relevance of a structure with correlated factors.

The present study has limitations that must be noted. A first limitation is related to our decision to allow the error terms from some items to intercorrelate, which is controversial. These correlation errors represent the common variance of the items that it is unexplained by the latent common factor. However, authors of previous studies have suggested that in the PANAS correlation errors are appropriate because theory and previous findings support them (Tuccitto, Giacobbi & Leite, 2009) and a reduced number of them are used (Crawford & Henry, 2004). It would be valuable to check whether the new correlation between the error terms of the items distressed and nervous, established in the model that has obtained the best fit to the data in the present study, could be replicated with data from other samples of the fibromyalgia population (MacCallum & Austin, 2000). A second limitation is lack of a representative sample of free-pain controls. Due to a limited budget, we were not able to recruit such a group to conduct further research into the factor invariance of affect structure between fibromyalgia participants (i.e., a clinical population) and the general population (Crawford & Henry, 2004). Similarly, we believe that it would be valuable to test this invariance across chronic pain conditions. Third, according to the dynamic model of affect, the correlation between Positive Affect and Negative Affect in a person with fibromyalgia could vary from time to time depending on his or her levels of stress induced by pain. The cross-sectional design of the present study did not allow us to performance more sophisticated analyses such as multifactor level analyses (Merz & Roesch, 2011). Also, experimental research addressing this question is warranted. Fourth, only women with fibromyalgia were included. Previous studies have showed that adult women are characterized by higher Negative Affect scores in both, state and trait measurements (Terracciano, McCrae & Costa, 2003). Thus, it could be interesting to assess the structural equivalence of Positive Affect and Negative Affect scores in men and women with fibromyalgia. The major difficulty to conduct this kind of studies in fibromyalgia is the small number of men with fibromyalgia, the reported women to men prevalence ratio is 9:1 (Branco et al., 2010). Fifth, this study was conducted in a sample of Spanish-speakers and, therefore, further research with other language versions of the PANAS would be valuable.

In conclusion, the present study demonstrates that both Positive Affect and Negative Affect are core dimensions of affect in adult women with fibromyalgia from Andalusia (Southern Spain). A structure with two correlated factors of the PANAS emerged from our sample of women with fibromyalgia. In this model, the amount of variance shared by Positive Affect and Negative Affect was small. Therefore, our findings support to use and interpret the Positive Affect and the Negative Affect subscales of the PANAS as separate factors that are associated but distinctive as well. It would be valuable to conduct further research into the invariance of the PANAS factor structure between people with fibromyalgia and pain-free peers, across chronic pain conditions, and between women and men with fibromyalgia. Additionally, experimental studies and multifactor level analyses addressing whether the correlation between Positive Affect and Negative Affect varies depending on current levels of stress induced by pain would be of interest.

Supplemental Information

Supplemental Information 1 Raw dataset.

Click here for additional data file.

Additional Information and Declarations

Competing Interests

Author Contributions

Human Ethics

Data Deposition

1 As Tuccitto, Giacobbi & Leite (2009) indicated, the items of the PANAS were taken from the mood subcategories (Zevon & Tellegen, 1982): attentive (e.g., interested, alert, attentive), excited (e.g., enthusiastic, inspired, excited), proud (e.g., determined, proud), strong (e.g., active, strong), distressed (e.g., upset, distressed), guilty (e.g., ashamed, guilty), angry (e.g., hostile, irritable), jittery (e.g., nervous, jittery), and fearful (e.g., scared, afraid).

The authors have no conflict of interest to disclose.

Fernando Estévez-López conceived and designed the experiments, performed the experiments, analyzed the data, contributed reagents/materials/analysis tools, wrote the paper, prepared figures and/or tables, reviewed drafts of the paper.

Manuel Pulido-Martos conceived and designed the experiments, performed the experiments, analyzed the data, contributed reagents/materials/analysis tools, wrote the paper, prepared figures and/or tables, reviewed drafts of the paper.

Christopher J. Armitage wrote the paper, reviewed drafts of the paper.

Alison Wearden wrote the paper, reviewed drafts of the paper.

Inmaculada C. Álvarez-Gallardo performed the experiments, contributed reagents/materials/analysis tools, reviewed drafts of the paper.

Manuel Javier Arrayás-Grajera performed the experiments, contributed reagents/materials/analysis tools, reviewed drafts of the paper.

María J. Girela-Rejón performed the experiments, contributed reagents/materials/analysis tools, reviewed drafts of the paper.

Ana Carbonell-Baeza conceived and designed the experiments, performed the experiments, contributed reagents/materials/analysis tools, reviewed drafts of the paper.

Virginia A. Aparicio conceived and designed the experiments, performed the experiments, contributed reagents/materials/analysis tools, reviewed drafts of the paper.

Rinie Geenen reviewed drafts of the paper.

Manuel Delgado-Fernández conceived and designed the experiments, performed the experiments, contributed reagents/materials/analysis tools, reviewed drafts of the paper.

Víctor Segura-Jiménez performed the experiments, contributed reagents/materials/analysis tools, reviewed drafts of the paper.

The following information was supplied relating to ethical approvals (i.e., approving body and any reference numbers):

The research protocol was reviewed and approved by the Ethics Committee of the Hospital Virgen de las Nieves (Granada, Spain); Registration number: 15/11/2013-N72.

The following information was supplied regarding data availability:

The raw data has been supplied as Supplemental Dataset Files.

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
