# Peer review of "Factor structure of the Positive and Negative Affect Schedule (PANAS) in adult women with fibromyalgia from Southern Spain: the al-Ándalus project"

_PeerJ, doi:10.7717/peerj.1822_

## Round 0.1 · original submission · Major Revisions

Dear Dr Estevez-Lopez

Thank you for submitting your work to PeerJ. I have had the privilege of two reviewers evaluating your manuscript in addition to my own evaluation. Please address all of the points that are raised by the reviewers. I particularly highlight some below:

1. Could you please explain your choice of method? Why SEM? Why not item-response theory?
2. Could you please discuss the full extent of your findings? For example, the feasibility of the correlated error terms.
3. Although this is about the factor structure of PANAS in fibromyalgia patients, there is little mention that this is also in Spanish. It should be clear in the title as well.

Reviewer 1 ·

Basic reporting

This article is a methodological study and exposes how authors assessed the structural construct validity of the PANAS in adult women with fibromyalgia using CFA.
This paper is elaborated in a clear, unambiguous and professional English used throughout. However, when the abstract describes: “the present study demonstrates positive affect and negative affect as core dimensions of affect in adult women in fibromyalgia” (lines 61-62); this description seems indefinite or incomplete. I recommend authors to specify this phrase. For example: “the present study demonstrates both positive affect and negative affect are core dimensions of affect in adult women in fibromyalgia” if authors would say this.
On the other hand, introduction and background show context clearly and literature is relevant, but not always well referenced:
There are some in-text citation that not fill norms of PeerJ:
In lines106-107 the in-text citation follow APA norms, ordered multiple references to the same item alphabetically; but PeerJ demands to order chronologically. The same problem occurs in the in-text citations en lines 132-133, 143, 152-153, 195, 246-48, 368-69, 392-93, 406-07,
The list of references meet the criteria of the APA, which does not contradict the requirements of PeerJ .
The structure is conform to PeerJ standard discipline norm.
Figures are relevant, high quality, well labelled & described.

Experimental design

This work is an original primary research within Scope of the journal.
Technical and ethical standard has been performed adequately.
The research question is well defined. In addition, this question is relevant and meaningful. It is stated how research fills an identified knowledge gap.
The authors assessed the structural construct validity of the PANAS in adult women with fibromyalgia using CFA. The association between the two dimensions of the PANAS in clinical populations is important for authors because it shows how the PANAS should be used and interpreted in clinical settings.
Most of the instruments created to assess the human suffering not always represented (in its data) the people who suffer, but also other kind of samples (student, i.e.). So, studies assessing outcomes in clinical populations are needed; as this article does.
Authors find a knowledge gap because their review of the literature elicited just two studies that have addressed the factor structure of the PANAS with CFA in clinical samples, but no one was with fibromyalgia, a disorder characterized by pain.
Methods are described with sufficient detail and information to replicate.

Validity of the findings

Impact and novelty
The main impact of the research is the suggestion that people with fibromyalgia may experience affect differently than healthy peers. The two factors of PANAS in the clinical sample are associate as well as distinctive, inversely to healthy women.

Data and statistical
Data is robust, statistically sound, & controlled. But some authors questioned the use of CFA or they proposed other options, as Exploratory Structural Equation Modeling (ESEM)*. I recommend authors to justify the methodological decision of apply CFA analysis over other methods, as it’s developed in this article.

Limitations of the study
Rightly, the authors notice the limitations of their study in the gender of the sample, justified in difficulties of prevalence of population with fibromyalgia. This limitation affect even to the title. Emotional results use to be different in men and in women but authors, appropriately, addressed the question at the end of the paper exposing the need of further research in affect structure between general population and clinical samples, as well among chronic pain conditions. Authors could propose an intention to project a study of this type or they could suggest a way to get it, overcoming the difficulties they found already.

* Lloret-Segura, Ferreres-Traver, Hernández-Baeza & Tomás-Marco. (2014). El análisis factorial exploratorio de los ítems: una guía práctica, revisada y actualizada. Anales de Psicología, 30(3), 1151-1169

·

Basic reporting

The basic reporting is OK, but there is some unusual phrasing that might be improved: "factor structure" is probably more commonly used than "factorial structure", for example; "two correlated factor structure" might be better phrased "structure with two correlated factors"; the Results section of the abstract starts with what was not found rather than what was found, which should lead.

The structure of the paper could be usefully re-worked as there are several related concepts explored and in the present draft they are inter-tangled. It seems to me that the paper is primarily concerned with the measurement properties (structure) of the PANAS and so these should be discussed in more detail in the introduction, with reference to previous findings (already done) but also the analysis methods used and how these might explain the differences reported (for example, I imagine some of these studies used EFA, some CFA, some SEM and these differences alone might explain the pattern of findings). Some of this material appears for the first time in the Methods section but is unlinked to the substantive discussion of the previous findings. A separate issue is how the construct of affect itself varies in different populations: this might explain the previous studies of the structure of the PANAS if the underlying constructs are different. Again, some reference is made to this but it needs to be clear how this material is related to the central question of the structure of the PANAS.

Experimental design

Nicely done, no problems here. I'm happy with the methods used but the choices should be justified - e.g. why use SEM and not item-response theory? Why choose those particular fit measures?

Alpha of 0.7 is not 'good' (it's an absolute minimum requirement). Also, the calculation of alpha assumes uncorrelated error terms so the authors should comment on this as their preferred model has correlated error terms.

Validity of the findings

I think there is too much discussion of the rejected/less good models which draws attention away from the key finding, i.e. the best model. The authors discuss the feasibility of the correlated error terms but not the implications for reliability (see comment above). The low correlation between the positive and negative factors is interesting.The authors discuss the findings thoroughly and list the many limitations.

Additional comments

I think the revisions required are quite minor but I would like to see the revised draft. This appears to make my Recommendation "Major Revisions" but the authors should not be unduly alarmed.

---

## Round 0.2 · accepted · Accept

Dear Dr Estevez-Lopez

Thank you for the revision of your manuscript. I am happy to let you know that we would like to accept it for publication.

·

Basic reporting

The authors have accommodated all my comments and I'm happy for this article to be accepted for publication.

Experimental design

The authors have accommodated all my comments and I'm happy for this article to be accepted for publication.

Validity of the findings

The authors have accommodated all my comments and I'm happy for this article to be accepted for publication.

Additional comments

The authors have accommodated all my comments and I'm happy for this article to be accepted for publication.